# EXTENDING MERCER'S EXPANSION TO INDEFINITE AND ASYMMETRIC KERNELS

**Sungwoo Jeong**
Department of Mathematics
Cornell University
sjeong@cornell.edu

**Alex Townsend**
Department of Mathematics
Cornell University
townsend@cornell.edu

## ABSTRACT

Mercer's expansion and Mercer's theorem are cornerstone results in kernel theory. While the classical Mercer's theorem only considers continuous symmetric positive definite kernels, analogous expansions are effective in practice for indefinite and asymmetric kernels. In this paper we extend Mercer's expansion to continuous kernels, providing a rigorous theoretical underpinning for indefinite and asymmetric kernels. We begin by demonstrating that Mercer's expansion may not be pointwise convergent for continuous indefinite kernels, before proving that the expansion of continuous kernels with bounded variation uniformly in each variable separably converges pointwise almost everywhere, almost uniformly, and unconditionally almost everywhere. We also describe an algorithm for computing Mercer's expansion for general kernels and give new decay bounds on its terms.

## 1 INTRODUCTION

Before the singular value decomposition (SVD) of a matrix, several pioneers of modern functional analysis in the early 20th century—Hammerstein (1923), Schmidt (1907), Smithies (1938), and Weyl (1912)—figured out the properties of the eigenfunction expansion and the singular value expansion (SVE) for a general square-integrable kernel (Stewart, 1993). Within these developments, Mercer, in 1909, showed that any continuous, symmetric positive definite kernel $K : [a, b] \times [a, b] \to \mathbb{R}$ with $-\infty < a < b < \infty$ can be expressed as an absolutely and uniformly convergent sum of orthonormal eigenfunctions multiplied by their corresponding eigenvalues, i.e.,

$$K(x, y) = \sum_{n=1}^{\infty} \lambda_n u_n(x) u_n(y), \quad (x, y) \in [a, b] \times [a, b]. \tag{1}$$

The expansion in eq. (1) is typically called a Mercer's expansion of $K$ (Mercer, 1909).

Mercer's expansion is an important result in functional analysis. It plays a crucial role in many machine learning applications because it underpins the theory of kernel methods (Scholkopf & Smola, 2002; Kung, 2014), allowing researchers to develop theoretical foundations for Gaussian processes (Kanagawa et al., 2018), support vector machines (SVM) (Cortes & Vapnik, 1995; Steinwart & Christmann, 2008), Transformer models (Vaswani et al., 2017; Wright & Gonzalez, 2021), and many other applications (Ghojogh et al., 2021).

Several researchers have been interested in using Mercer-like expansions for continuous kernels that are symmetric but not positive definite (indefinite kernels) as well as general continuous kernels that are not symmetric (asymmetric kernels). There are at least three reasons why indefinite and asymmetric kernels are useful: (1) Checking whether a kernel is positive definite can be computationally expensive (Luss & d'Aspremont, 2007; Huang et al., 2017), (2) Often one finds that Mercer-like expansions continue to hold for indefinite and asymmetric kernels, even though the theoretical underpinnings have been lacking, and (3) Indefinite and asymmetric kernels, such as the sigmoid (tanh) kernel, naturally arise in applications such as Transformers (Wright & Gonzalez, 2021; Chen et al., 2024), kernel learning theory (Haasdonk & Keysers, 2002; Ong et al., 2004; He et al., 2023), $H^\infty$ control theory (Hassibi et al., 1999), protein sequence similarity (Saigo et al., 2004; Vert et al., 2004), and many others.

Surprisingly, a continuous symmetric kernel may not have a Mercer-like expansion that converges pointwise (see Proposition 1). We hope this new example clarifies some confusion in the literature on Mercer's expansion for general kernels. We also prove that general continuous kernels may also have Mercer's expansions that converge pointwise, but not uniformly and absolutely, meaning that Mercer's expansion must be used with some care for general kernels.

This work provides a rigorous theoretical underpinning of Mercer's expansion for indefinite and asymmetric kernels. For indefinite kernels, the eigenfunction expansion is the most natural generalization of Mercer's expansion. For asymmetric kernels, the most natural generalization is the following singular value expansion (SVE) that was considered in the early 1900s (Schmidt, 1907)

$$K(x,y) = \sum_{n=1}^{\infty} \sigma_n u_n(x) v_n(y), \tag{2}$$

where $\sigma_1 \geq \sigma_2 \geq \cdots > 0$ are the singular values of $K$ and $\{u_n\}_{n\in\mathbb{N}}$ and $\{v_n\}_{n\in\mathbb{N}}$ are sets of orthonormal functions called the right and left singular functions of $K$, respectively. The expansion in eq. (2) can replace Mercer's expansion for asymmetric kernels (see Section 3). We regard eq. (2) as a natural generalization of Mercer's expansion to general continuous kernels.

Schmidt proved the existence of the expansion in eq. (2) for all square-integrable kernels and showed that the equality holds in the $L^2$ sense (Schmidt, 1907). He also proved that for continuous kernels $K$, if $\sigma_n > 0$, the corresponding left and right singular functions can be selected as continuous. Later, in 1923, Hammerstein showed that the expansion in eq. (2) converges uniformly for any continuous kernel that also satisfies a contrived square-integrated Lipschitz-like condition (Hammerstein, 1923). Later, in 1938, Smithies gave a finicky condition on continuous kernel that ensures specific decay rates of the singular values, and he used that decay to prove almost everywhere pointwise and almost everywhere absolute convergence (Smithies, 1938). More recently, it was realized that both Hammerstein's and Smithies' conditions are satisfied by continuous kernels that are Lipschitz continuous uniformly in each variable separably (Townsend, 2014).

We prove results for a larger class of general continuous kernels $K : [a,b] \times [c,d] \to \mathbb{R}$ than Lipschitz continuous kernels. We only need the kernels to have bounded variation uniformly in each variable separately, i.e., for any fixed $x \in [a,b]$ and any fixed $y \in [c,d]$, we have

$$\int_a^b \frac{\partial}{\partial x} K(x,y) dx < V, \qquad \int_c^d \frac{\partial}{\partial y} K(x,y) dy < V, \tag{3}$$

for some fixed uniform constant $V < \infty$. We say a kernel is of uniform bounded variation if it satisfies the conditions in eq. (3). The partial derivatives in eq. (3) are replaced by the notion of weak derivatives if the kernel is not differentiable (Trefethen, 2019, Chap. 7). We also provide new decay bounds on $\sigma_n$ for smoother kernels (see Section 4), which we believe are asymptotically tight, and can be used to rigorously truncate Mercer's expansion.

We prove a new result that continuous kernels of uniform bounded variation have a Mercer's expansion that converges pointwise almost everywhere, unconditionally almost everywhere, and almost uniformly. We have not been able to prove that the expansion converges pointwise, absolutely, and uniformly, so there could still be some subtleties in using Mercer's expansion for general kernels. However, we believe that continuous kernels of uniform bounded variation have a Mercer's expansion that converges pointwise, absolutely, and uniformly, and we hope later research will prove this.

## 1.1 OUR CONTRIBUTIONS

We generalize Mercer's expansion to a general continuous kernel, where the kernel may be indefinite or asymmetric. The following two theorems summarize our main theoretical results about Mercer's expansion for general continuous kernels (see Section 3).

**Theorem.** *For any $[a,b] \subset \mathbb{R}$ there are continuous symmetric indefinite kernels on $[a,b] \times [a,b]$ such that Mercer's expansion in eq.* (2)*: (i) does not converge pointwise, (ii) converges pointwise but not uniformly, and (iii) converges pointwise but not absolutely.*

**Theorem.** *For any $[a,b], [c,d] \subset \mathbb{R}$, let $K : [a,b] \times [c,d] \to \mathbb{R}$ be a continuous kernel of uniform bounded variation (see eq.* (3)*). Then, Mercer's expansion of $K$ in eq.* (2) *converges pointwise, unconditionally almost everywhere, and almost uniformly.*

The first theorem shows us that Mercer's expansion for continuous kernel must be treated with significant care. However, our result resolve some of these subtleties for continuous kernels of uniform bounded variation. Bounded variation is a weak extra smoothness condition on a continuous kernel, e.g., any uniform Lipschitz continuous kernel (in each variable separably) is also of uniform bounded variation. In addition to these two main theorems, we prove new decay bounds on the singular values $\sigma_n$ for smoother kernels (see Section 4). We also describe an efficient algorithm to compute Mercer's expansion in eq. (2) for general continuous kernels (see Section 5).

## 2 BACKGROUND: MERCER'S THEOREM FOR GENERAL KERNELS

In this section, we provide some background material on Mercer's theorem and review some of the literature's efforts to extend Mercer's expansion to general kernels.

### 2.1 MERCER'S THEOREM FOR CONTINUOUS SYMMETRIC POSITIVE DEFINITE KERNELS

Given a continuous symmetric positive definite kernel $K$, Mercer's theorem ensures the pointwise, uniform, and absolute convergence of a Mercer's expansion of $K$ (Mercer, 1909).

**Theorem 1** (Mercer's theorem). *For any $[a, b] \subset \mathbb{R}$, let $K : [a, b] \times [a, b] \to \mathbb{R}$ be a continuous symmetric function. Suppose $K$ is positive definite, i.e., $\sum_{i,j=1}^{N} c_i c_j K(x_i, x_j) \geq 0$ for any $c_1, \ldots, c_N \in \mathbb{R}$ and $x_1, \ldots, x_N \in [a, b]$. Then, there is a set of continuous orthonormal functions $\{u_n\}_{n \in \mathbb{N}}$ on $[a, b]$ and corresponding positive real numbers $\lambda_1 \geq \lambda_2 \geq \cdots > 0$ such that*

$$\int_a^b K(x, y) u_n(y) dy = \lambda_n u_n(x), \quad x \in [a, b].$$

*Moreover, $K$ has a series expansion given by*

$$K(x, y) = \sum_{n=1}^{\infty} \lambda_n u_n(x) u_n(y), \quad x, y \in [a, b], \tag{4}$$

*where the series converges pointwise, uniformly, and absolutely to $K$.*

Mercer's theorem is a simple but powerful result showing us that Mercer's expansion for continuous symmetric positive definite kernels can be used without any subtleties from an analysis perspective. In this setting, by spectral theory, $\lambda_1, \lambda_2, \ldots$, are the positive eigenvalues of $K$ with corresponding orthonormal eigenfunctions $\{u_n\}_{n \in \mathbb{N}}$.

The convergence properties of Mercer's expansion are useful in integral operator theory (Stewart, 2011). For instance, if a Mercer's expansion of $K : [a, b] \times [a, b] \to \mathbb{R}$ converges uniformly, one can interchange integration and the summation, i.e.,

$$\int_a^b K(x, y) f(y) dy = \int_a^b \sum_{n=1}^{\infty} \sigma_n u_n(x) v_n(y) f(y) dy = \sum_{n=1}^{\infty} \sigma_n u_n(x) \int_a^b v_n(y) f(y) dy.$$

In particular, the integrals $\int v_n(y) f(y) dy$ do not depend on $x$, so $\int_a^b K(x, y) f(y) dy$ can be computed efficiently using this formula. Moreover, the integral equation $g(x) = \int K(x, y) f(y) dy$, with $f$ unknown, can be easily solved by back substitution using Mercer's expansion (Schmidt, 1907).

When Mercer's expansion converges uniformly, its truncation error is always bounded in the uniform norm. This means that one can safely truncate the series and use finite rank approximations of the kernel. On the other hand, if Mercer's expansion does not converge uniformly, the uniform norm error between $K$ and a truncation may not converge to zero as the rank increases (see Section 3.3).

Finally, when Mercer's expansion converges absolutely (or unconditionally), one can write the kernel in factored form, i.e., $K = U\Sigma V^{\top}$. Writing a function in a decomposition format like this has several advantages, as linear algebra algorithms immediately extend to the continuous setting (see (Townsend & Trefethen, 2015)). Our result in Section 3.4 implies a similar but weaker conclusion.

## 2.2 PREVIOUS WORK ON INDEFINITE AND ASYMMETRIC KERNELS

There are several attempts to establish a theoretical framework for Mercer's expansions of indefinite and asymmetric kernels. One of the most popular attempts is the theory of Reproducing Kernel Kreĭn Space (RKKS) (Alpay, 2001; Ong et al., 2004; Loosli et al., 2015; Huang et al., 2017) for indefinite kernels. Here, a kernel $K$ needs to have a so-called positive decomposition $K(x, y) = K_+(x, y) - K_-(x, y)$, where $K_+$ and $K_-$ are continuous symmetric positive definite kernels (which is also required for the generalized Mercer's theorem in (Chen et al., 2008)). Another attempt is the Reproducing Kernel Banach Space (RKBS), which removes the inner-product requirement from an RKHS. It can be defined in several ways (Zhang et al., 2009; Georgiev et al., 2013; Xu & Ye, 2019; Lin et al., 2022), for example, the RKBS requires a semi-inner product in (Zhang et al., 2009) and a $L^p$-norm in (Georgiev et al., 2013).

Unfortunately, not all continuous indefinite kernels induce an RKKS, and not all continuous kernels induce an RKBS. Characterizing what properties a kernel must have to induce an RKKS or RKBS remains an open problem. For example, the continuous indefinite kernel $K_{abs}$ in eq. (6) does not have a positive decomposition, so it does not define an RKKS. Likewise, there are continuous kernels for which Mercer's expansion does not converge pointwise (see Proposition 1) and hence does not satisfy the conditions in (Xu & Ye, 2019) to generate an associated RKBS.

Aside from reproducing kernel approaches, researchers have tried to generate feature spaces induced by indefinite kernels. Using pseudo-Euclidean spaces, (Haasdonk, 2005) shows that SVMs are a sort of distance minimizer in such a space, although it is not a metric space. Other studies handle indefinite kernel learning by modifying problems to continuous symmetric positive definite kernel learning either by transforming the kernels themselves (Wu et al., 2005) or by finding a nearby positive definite kernel (Luss & d'Aspremont, 2007). In fact, such modification may not be possible for some kernels (see, for example, proofs of Propositions 2, 3) thus careful rigorous analysis has to be entailed. We hope the theoretical underpinnings detailed in this paper can aid future attempts in these directions.

## 3 MERCER'S EXPANSION FOR GENERAL CONTINUOUS KERNELS

We detail Mercer's expansion for continuous kernels and investigate its convergence behavior.

### 3.1 MERCER'S EXPANSION

Consider a general continuous kernel $K : [a, b] \times [c, d] \to \mathbb{R}$, where $[a, b], [c, d] \subset \mathbb{R}$. If $K$ is symmetric, then the interval $[c, d]$ will equal $[a, b]$. As hinted by the SVD for matrices, for a general kernel $K$, we will need to consider two sets of functions: (1) Right singular functions denoted by $u_1, u_2, \ldots$, which are orthonormal with respect to $L^2([a, b])$ and (2) Left singular functions denoted by $v_1, v_2, \ldots$, which are orthonormal with respect to $L^2([c, d])$. The functions are defined to satisfy

$$\sigma_n u_n(x) = \int_c^d K(x, y) v_n(y) dy, \qquad \sigma_n v_n(y) = \int_a^b K(x, y) u_n(x) dx. \qquad (5)$$

The values $\sigma_1 \geq \sigma_2 \geq \cdots > 0$ are called the positive singular values of $K$. The SVE of $K$, which we refer to as Mercer's expansion, is given in eq. (2).[1]

The SVE of $K$ is not completely unique. However, the convergence properties are the same for any SVE of $K$. One can modify the left and right singular functions with signs and potentially reindex terms when singular values plateau. Moreover, the integral definition in eq. (5) allows the left and right singular functions to be modified on sets of measure zero. For continuous kernels, if $\sigma_n > 0$, we select $u_n$ and $v_n$ to be continuous, which Schmidt (1907) showed is always possible. With this convention, the SVE of $K$ is equivalent to Mercer's expansion for continuous symmetric positive definite kernels.

---

[1] If $K$ is continuous, then the equality at least holds in the $L^2$ sense.

## 3.2 MERCER'S EXPANSION MAY NOT CONVERGE POINTIWSE

Given a continuous symmetric but indefinite kernel $K : [a, b] \times [a, b] \to \mathbb{R}$, a Mercer's expansion in eq. (2) may not converge pointwise.

**Proposition 1.** *For any $[a, b] \subset \mathbb{R}$, there exists a continuous symmetric indefinite kernel $K_{pt} : [a, b] \times [a, b] \to \mathbb{R}$ with a Mercer's expansion that does not converge pointwise.*

*Proof.* We prove this by providing an example and, without loss of generality, we assume that $[a, b] = [-1, 1]$. Consider a continuous even $2\pi$-periodic function $f : [-\pi, \pi] \to \mathbb{R}$ whose Fourier series does not converge pointwise at $t = 0$. Such a function exists, for example, $f(t) = \sum_{n=1}^{\infty} \sin(2^{n^3} + 1)t/2)/n^2$ (Fejér, 1911). Let $f(t) = \sum_{n=-\infty}^{\infty} c_n e^{int} = c_0 + \sum_{n=1}^{\infty} 2c_n \cos(nt)$ be the Fourier expansion of $f(t)$, i.e., $c_n = \frac{1}{2\pi} \int_{-\pi}^{\pi} f(t) e^{int} dt$. Now, consider $K_{\text{pt}}(x, y) := f(x - y)$. Since $f$ is $2\pi$-periodic we find that for fixed $y$, we have $\int_{-\pi}^{\pi} K_{\text{pt}}(x, y) e^{inx} dx = 2\pi c_n e^{iny}$ for $n \in \mathbb{Z}$. We also have $\int_{-\pi}^{\pi} K_{\text{pt}}(x, y) \cos(nx) dx = 2\pi c_n \cos(ny)$ and $\int_{-\pi}^{\pi} K_{\text{pt}}(x, y) \sin(nx) dx = 0$ by Euler's formula. Since $\{\cos nx\}_{n \in \mathbb{N}} \cup \{\sin nx\}_{n \in \mathbb{N}}$ form a complete orthogonal basis of $L^2([-\pi, \pi])$, we have the full set of eigenfunctions of $K_{\text{pt}}$. Thus,

$$K_{\text{pt}}(x, y) = c_0 + \sum_{n=1}^{\infty} 2\pi c_n \frac{1}{\sqrt{\pi}} \cos(nx) \frac{1}{\sqrt{\pi}} \cos(ny) = c_0 + \sum_{n=1}^{\infty} 2c_n \cos(nx) \cos(ny),$$

where equality holds in the $L^2$ sense. To conclude that the series does not converge pointwise, consider the expansion at $y = 0$. We find that $K_{\text{pt}}(x, 0) = c_0 + \sum_{n=1}^{\infty} 2c_n \cos(nx) = f(x)$. Since this is the Fourier series of $f$, we know that this series does not converge pointwise at $0$. We conclude that we have a Mercer's expansion for $K_{\text{pt}}$ that is not pointwise convergent at $(0, 0)$. This example can be transplanted to have a domain $[a, b] \times [a, b]$ by a linear translation for any $[a, b] \subset \mathbb{R}$. □

Since a general kernel may be indefinite, the example in the proof of Proposition 1 also shows that Mercer's expansion may not converge pointwise for general kernels.

## 3.3 MERCER'S EXPANSION MAY NOT CONVERGE ABSOLUTELY OR UNIFORMLY

For a continuous kernel, Mercer's expansion might converge pointwise but not absolutely or might converge pointwise but not uniformly. One must treat Mercer's expansion carefully for indefinite or asymmetric continuous kernels because its convergence properties can be subtle.

**Proposition 2.** *For any $[a, b] \subset \mathbb{R}$ there exists a continuous symmetric indefinite kernel $K_{abs} : [a, b] \times [a, b] \to \mathbb{R}$ with a Mercer's expansion that converges pointwise but not absolutely.*

**Proposition 3.** *For any $[a, b] \subset \mathbb{R}$ there exists a continuous symmetric indefinite kernel $K_{uni} : [a, b] \times [a, b] \to \mathbb{R}$ with a Mercer's expansion that converges pointwise but not uniformly.*

We prove these two propositions in Appendix A by finding kernels that possess Mercer's expansion with the desired properties. If a kernel has only a finite number of negative (positive) eigenvalues, one can subtract a finite rank function from $K$ to make it positive (negative) definite. This allows one to use the classic Mercer's theorem for continuous positive definite kernels, ensuring that Mercer's expansion converges pointwise, absolutely, and uniformly. Thus, the kernels we search for must have an infinite number of both positive and negative eigenvalues.

For example, as a proof of Proposition 2, we show that the following continuous kernel $K_{\text{abs}} : [-1, 1] \times [-1, 1] \to \mathbb{R}$ has a Mercer's expansion that converges pointwise but not absolutely:

$$K_{\text{abs}}(x, y) = \sum_{n=1}^{\infty} \frac{(-1)^n}{n^2} \tilde{P}_{2n}(x) \tilde{P}_{2n}(y), \tag{6}$$

where $\tilde{P}_n$ is the degree $n$ normalized Legendre polynomial (in the $L^2$ sense), i.e., $\tilde{P}_n(x) = \sqrt{\frac{2n+1}{2}} P_n(x)$. The alternating signs in $(-1)^n$ in eq. (6) are essential to create a kernel with infinitely many positive and infinitely many negative eigenvalues. See Appendix A for details.

## 3.4 ON THE CONVERGENCE OF MERCER'S EXPANSION

We now prove new convergence results on Mercer's expansion of continuous kernels of uniform bounded variation (see eq. (3) for definition). Intuitively, a continuous function with bounded variation does not have infinitely many oscillations with non-negligible variation. Almost all continuous functions that appear in practice are of uniform bounded variation. In particular, we believe that all the continuous kernels used in machine learning are of uniform bounded variation. Therefore, our extra smoothness requirement on continuous kernels is a technical assumption needed for the analysis but not a restrictive condition in practice.

We also remark that while Mercer's theorem holds for any compact domain, we only consider closed subsets of $\mathbb{R}$ in this work. However, in any case, Propositions 1 to 3 still serve as counterexamples for Mercer's theorem for general kernels on general domains. Mercer's theorem is special since the series has small amount of cancellation, thus the convergence does not rely on properties of orthonormal functions. However for general domains cancellation of orthonormal series has to be carefully examined, which is a relatively less-known subject except for closed subsets of $\mathbb{R}$.

### 3.4.1 UNCONDITIONALLY ALMOST EVERYWHERE CONVERGENCE

We first prove a decay rate on the singular values. We use a recent result in (Wang, 2023) to obtain a bound of the coefficients in a Legendre expansion and then use Eckart–Young–Mirsky theorem for the SVE (Schmidt, 1907; Eckart & Young, 1936; Mirsky, 1960) to conclude Proposition 4.

**Proposition 4.** *For any $[a, b], [c, d] \subset \mathbb{R}$, a continuous kernel $K : [a, b] \times [c, d] \to \mathbb{R}$ of uniform bounded variation (see eq. (3)) has $\sigma_n = \mathcal{O}(n^{-1})$ as $n \to \infty$.*

A proof and discussion of Proposition 4 is given in Section 4. Proposition 4 tells us that the extra smoothness assumption on $K$ leads to a faster decay of the singular values. For a general continuous kernel $K : [a, b] \times [c, d] \to \mathbb{R}$, we can conclude that it is square-integrable so that $\sum_{n=1}^{\infty} \sigma_n^2 < \infty$. This only allows us to conclude that $\sigma_n = \mathcal{O}(n^{-1/2})$, not $\sigma_n = \mathcal{O}(n^{-1})$. The extra decay of the singular values allows us to show that Mercer's expansion convergence unconditionally almost everywhere. We prove a more general result (see Appendix B for the proof).

**Theorem 2** (Unconditional Convergence of Mercer's expansion)**.** *For any $[a, b], [c, d] \subset \mathbb{R}$ and a continuous kernel $K : [a, b] \times [c, d] \to \mathbb{R}$ with singular values satisfying $\sigma_n \leq Cn^{-\alpha}$ for some $\alpha > \frac{1}{2}$ and some $C > 0$, a Mercer's expansion of $K$ converges unconditionally almost everywhere.*

To prove Theorem 2, we use the theory of orthonormal series (see Lemma 3), which can be thought of as an extension of the Rademacher–Menchov theorem. We immediately conclude the following result by combining Proposition 4 and Theorem 2.

**Corollary 1.** *Let $[a, b], [c, d] \subset \mathbb{R}$. Any continuous kernel $K : [a, b] \times [c, d] \to \mathbb{R}$ of uniform bounded variation (see eq. (3)) has a Mercer's expansion that converges unconditionally almost everywhere.*

Corollary 1 tells us that continuous kernels of uniform bounded variation have a Mercer's expansion that converges almost everywhere, regardless of the order in which the terms are summed. Unconditional almost everywhere convergence is closely related to absolute convergence. Unconditional almost everywhere convergence ensures that the expansion's convergence behavior is stable across the domain and that almost every point adheres to the same convergence properties. It is weaker than absolute convergence because there is possibly an exception set of measure zero.

One can also use Theorem 2 to conclude that Mercer's expansions converge unconditionally almost everywhere under alternative smoothness assumptions. For example, all continuous kernels that are Hölder continuous uniformly in each variable separately, i.e., there is a constant $C < \infty$ and $\gamma > 0$ such that

$$|K(x_1, y) - K(x_2, y)| \leq C|x_1 - x_2|^{\gamma}, \quad |K(x, y_1) - K(x, y_2)| \leq C|y_1 - y_2|^{\gamma},$$

for all $x_1, x_2, y_1, y_2, x, y$, have a Mercer's expansion that converges unconditionally almost everywhere. This is because such kernels have the singular value decay $\sigma_n \leq Cn^{-\frac{1}{2}-\gamma}$, which can be proved by Theorem 12 of (Smithies, 1938) with the choice of $p = 2$.

### 3.4.2 ALMOST UNIFORM CONVERGENCE

We now consider the uniform convergence of Mercer's expansion. We show that the expansion converges almost uniformly for continuous kernels with uniform bounded variation.

**Definition 1.** *A sequence of functions $\{f_n\}_{n\in\mathbb{N}}$ is called almost uniformly convergent on a measurable set $E$ if for any $\epsilon > 0$ there exists a set $E_\epsilon$ with measure smaller than $\epsilon$, such that the sequence $\{f_n\}_{n\in\mathbb{N}}$ converges uniformly on $E\backslash E_\epsilon$.*

We obtain our second main convergence result as follows.

**Theorem 3.** *With the same assumptions as Corollary 1, Mercer's expansion of $K$ converges almost uniformly in $y$ for almost every (fixed) $x$, and it converges almost uniformly in $x$ for almost every $y$.*

*Proof.* From Corollary 1, for almost every (fixed) $x$, we have the unconditional convergence of Mercer's expansion for almost every $y$. Since unconditional convergence implies pointwise convergence, Egorov's theorem (Egoroff, 1911) tells us that Mercer's expansion converges almost uniformly for $y$, for almost every $x$. The two-dimensional almost uniform convergence property also follows from the fact that Mercer's expansion converges pointwisely for almost every $(x, y) \in [a, b] \times [c, d]$. $\qquad\square$

The distinction between almost uniform convergence and uniform convergence is subtle. Uniform convergence means that the expansion converges to a limit so that the convergence rate is the same across the entire domain. In contrast, almost uniform convergence is a relaxation by allowing exceptional sets of arbitrarily small measure.

## 4 DECAY OF SINGULAR VALUES FOR SMOOTH KERNELS

Mercer's expansion is equivalent to the SVE of a kernel, and this means that if Mercer's expansion for $K$ is truncated after $k$ terms, then it forms a rank $k$ kernel that is a best rank-$k$ approximation[2] to $K$ in the $L^2$ sense (Schmidt, 1907). Thus, singular value bounds help understand the approximation power of low-rank approximations (Reade, 1983; Townsend, 2014).

Recall that $\tilde{P}_n(x)$ is normalized Legendre polynomial of degree $n$. If $f : [-1, 1] \to \mathbb{R}$ is a continuous function of bounded variation, then its Legendre expansion is given by

$$f(x) = \sum_{n=0}^{\infty} a_n \tilde{P}_n(x), \quad a_n = \int_{-1}^{1} f(x)\tilde{P}_n(x)dx,$$

which converges absolutely and uniformly to $f$ (Wang, 2023). Moreover, the $L^2$ projection of $f$ onto the space of polynomials of degree $\leq k$ is precisely the truncation of this series, i.e., $f_k(x) = \sum_{n=0}^{k} a_n \tilde{P}_n(x)$. When $f$ is $(r-1)$-times continuously differentiable and its $r^{\text{th}}$ derivative, $f^{(r)}$, has bounded variation, $V < \infty$, then for any $k > r + 1$, it holds that (Wang, 2023, Thm 3.5)

$$\|f - f_{k-1}\|_{L^2} \leq \frac{\sqrt{2}V}{\sqrt{\pi(r+\frac{1}{2})}(k-r-1)^{r+\frac{1}{2}}} = \mathcal{O}(k^{-(r+\frac{1}{2})}). \tag{7}$$

One can use eq. (7) to bound the singular values of kernels defined on $[-1, 1] \times [-1, 1]$ using an analogous argument to that found in (Reade, 1983). If $K(x, \cdot)$ is $(r-1)$-times continuously differentiable and its $r^{\text{th}}$ derivative in $x$ has bounded variation, $V < \infty$, uniformly in $y$,[3] then the Eckart–Young–Mirsky theorem shows that $K_k^{\text{leg}}(x, y) = \sum_{n=0}^{k-1} a_n(y)\tilde{P}_n(x)$ satisfies

$$\sqrt{\sum_{n=k+1}^{\infty} \sigma_n^2} \leq \|K - K_k^{\text{leg}}\|_{L^2} \leq \max_{y\in[-1,1]} \sqrt{2}\|K(\cdot, y) - \sum_{n=0}^{k-1} a_n(y)\tilde{P}_n(\cdot)\|_{L^2} \leq \frac{2V(\pi(r+\frac{1}{2}))^{-\frac{1}{2}}}{(k-r-1)^{r+\frac{1}{2}}},$$

---

[2]A nonzero kernel $K$ is rank-1 if it can be written as $K(x, y) = g(x)h(y)$. A sum of $k$ rank-1 kernels is of rank $\leq k$.

[3]Recall that we say that $K(x, \cdot)$ is of bounded variation uniformly in $y$ if the function $x \to K(x, y)$ is of bounded variation for every $y$ with the same constant $V$.

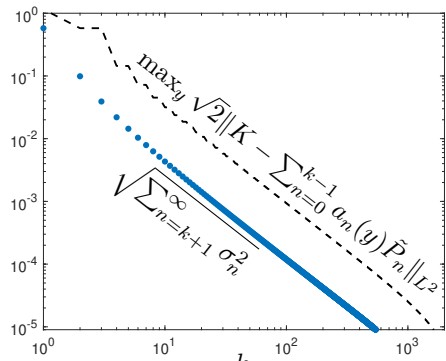 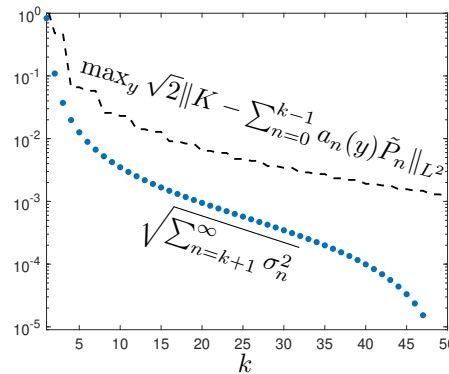

Figure 1: Singular value decay of two kernels defined on $[-1,1] \times [-1,1]$: (Left) $K(x,y) = \max\{0, 1 - |x| - |y|\}$ and (Right) $K(x,y) = \exp(-(x^2 + y^2)/200)\max(0, (1 - |x| - |y|))$. The bounds given by the truncated Legendre expansions can be tight, allowing one to determine the number of terms required in Mercer's expansion to achieve a desired accuracy.

where the first inequality comes from the fact that $\sqrt{\sum_{n=k+1}^{\infty} \sigma_n^2}$ is the best rank-$k$ approximation error to $K$ and the last inequality comes from eq. (7).

Since the singular values are indexed in non-increasing order, we have $k\sigma_{2k}^2 \le \sum_{n=k+1}^{2k} \sigma_n^2 \le \sum_{n=k+1}^{\infty} \sigma_n^2$. Thus, for $k > r+1$, we have

$$\sigma_{2k} \le \frac{2V(\pi(r+\frac{1}{2}))^{-\frac{1}{2}}}{\sqrt{k}(k-r-1)^{r+\frac{1}{2}}} = \mathcal{O}(k^{-(r+1)}). \tag{8}$$

The same bound on $\sigma_n$ holds if $K(\cdot, y)$ is $(r-1)$-times continuously differentiable and its $r^{\text{th}}$ derivative in $y$ has bounded variation $V$, uniformly in $x$. By transplanting kernels defined on $[a,b] \times [c,d]$, the bounds in eq. (8) can be modified for kernels defined on any domain. We believe the bounds in eq. (8) are asymptotically tight. A reader can refer to Figure 2.4 (right) of (Townsend, 2014) for related numerical experiments.

When $K$ is continuous with uniform bounded variation, we have $r = 0$ and eq. (8) shows that $\sigma_n = \mathcal{O}(n^{-1})$, which proves Proposition 4. The bounds in eq. (8) show us that the smoother a kernel, the faster its singular values decay to zero. In practice, if a kernel has rapidly decaying singular values its Mercer's expansion can be truncated after a small number of terms.

## 5 COMPUTING MERCER'S EXPANSION FOR GENERAL KERNELS

We now describe an algorithm for computing a Mercer's expansion of a kernel, which is described in (Townsend & Trefethen, 2013) for computing low-rank function approximations. Here, we use the fact that it is observed to compute near-best low-rank function approximations (Townsend, 2014). It involves two main steps: (1) Forming a low-rank approximant using a pseudo-skeleton approximation and (2) Compressing the approximant using a low-rank SVD (Bebendorf, 2008, Chapt. 1.1.4). This procedure can be much more efficient than sampling the kernel on a large grid and computing the matrix SVD. An algorithmic summary (and discussion on its complexity) can also be found in Figure 6.1 of (Townsend & Trefethen, 2013).

### 5.1 STEP 1: COMPUTING A PSEUDO-SKELETON APPROXIMANT

In the first step, we compute a pseudo-skeleton approximation using Gaussian elimination with complete pivoting (GECP), which is an iterative procedure to approximate the kernel $K(x,y)$ as a sum of rank-1 functions (Townsend & Trefethen, 2013). The algorithm is implemented in the two-dimensional side of Chebfun (Driscoll et al., 2014). Similar algorithms are called Adaptive Cross Approximation (Bebendorf, 2000) and maximum volume pseudo-skeleton approximation (Goreinov et al., 1997).

 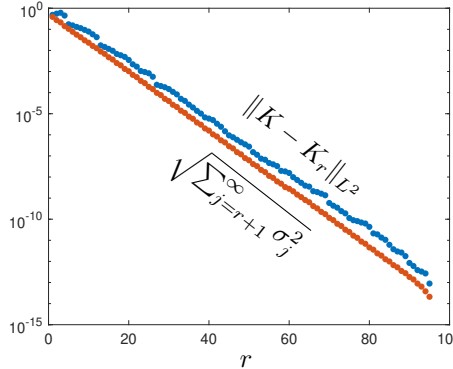

Figure 2: The kernel $K(x, y) = \tanh(100xy+1)$ on $[-1, 1] \times [-1, 1]$. Left: Step 1 of our algorithm selects $R = 96$ pivot locations (red dots). The algorithm densely samples $K$ along $2R$ lines (blue lines) to form its pseudoskeleton approximant. Right: The best $L^2$ error compared to $\|K - K_r\|_{L^2}$ for $1 \le r \le 96$, where $K_r$ is the rank $\le r$ pseudoskeleton approximation of $K$. At $R = 96$, step 1 of our algorithm terminates as it has pointwise approximated $K$ to extremely high accuracy.

The algorithm starts by defining $e_0 = K$ and finding an approximate location of a maximum absolute value of $e_0(x, y)$. A rank-1 function approximation is constructed that interpolates $K$ along the $x = x_0$ and $y = y_0$ lines, i.e.,

$$K_1(x, y) = \frac{e_0(x_0, y)e_0(x, y_0)}{e_0(x_0, y_0)}.$$

Then, we define the residual $e_1 = K - K_1$, and the process is repeated on $e_1$ to form $e_2$, and so on. After $R$ steps, a rank-$R$ approximation is constructed that interpolates $K$ along $2R$ lines. We stop the iteration when the residual error is below a user-defined tolerance such as $10^{-5}$.

To find an approximate location of a maximum absolute value, i.e., the pivot location, at each step of the algorithm, we sample $e_j$ for $j = 0, 1, \ldots$, on a coarse Chebyshev tensor grid of an adaptively selected size and select an absolute maximum from that grid. In this way, we only need to evaluate the kernel $K$ on coarse grids, and $K$ does not need to be densely sampled. At the end of step 1, we have constructed a rank $\le R$ approximation, given by

$$K_R(x, y) = \sum_{j=1}^{R} c_j \phi_j(x) \psi_j(y) \tag{9}$$

where $c_j = 1/e_{j-1}(x_{j-1}, y_{j-1})$ are the reciprocals of the pivot values, and $\phi_j(x) = e_{j-1}(x, y_{j-1})$ and $\psi_j(y) = e_{j-1}(x_{j-1}, y)$ are univariate functions. We represent $\phi_j$ and $\psi_j$ as Chebyshev expansions of an adaptively selected degree, which requires that $K$ is densely sampled along $2R$ lines.

This first step is observed to compute a near-optimal rank $\le R$ approximation to kernels. In Figure 2 we use it to approximate $K(x, y) = \tanh(100\,xy + 1)$ on the domain $[-1, 1] \times [-1, 1]$, where it decides that $R = 96$ is sufficient to approximation $K$ to extremely high precision. More algorithmic details and experiments are given in (Townsend & Trefethen, 2013; Townsend, 2014).

## 5.2 STEP 2: LOW-RANK SVD OF THE PSEUDO-SKELETON APPROXIMANT

After we have computed a rank $\le R$ pseudo-skeleton approximation in step 1, we improve it by performing a low-rank SVD. The SVD decomposes $K_R(x, y)$ into a sum of outer products of orthonormal functions with singular values and gives us an accurate truncated Mercer's expansion of $K$. To do this, we write in the form $K_R(x, y) = \Phi(x)C\Psi(y)^\top$, where

$$\Phi(x) = [\phi_1(x) \quad \cdots \quad \phi_R(x)], \quad \Psi(x) = [\psi_1(y) \quad \cdots \quad \psi_R(y)], \quad C = \begin{bmatrix} c_1 & & \\ & \ddots & \\ & & c_R \end{bmatrix}.$$

The procedure for computing the low-rank SVD of $K_R$ involves the following steps, which is essentially a fast way to compute a Mercer's expansion of a finite rank kernel:

1. Perform two QR decompositions of $\Phi(x)$ and $\Psi(y)$, using a function analogue of Householder QR (Trefethen, 2010). We can write this QR decomposition as $\Phi(x) = Q^{\text{left}}(x)R_1$ and $\Psi(y) = Q^{\text{right}}(y)R_2$, where $R_1, R_2 \in \mathbb{R}^{R \times R}$ and the columns of $Q^{\text{left}}(x)$ and $Q^{\text{right}}(y)$ are orthonormal functions.
2. Compute the SVD of an $R \times R$ matrix formed by $R_1 C R_2^\top = U\Sigma V^\top$.
3. Construct the final SVD-based approximation by combining the singular values and the orthonormalized functions to form:

$$K_R(x, y) = \sum_{j=1}^{R} \sigma_j u_j(x) v_j(y), \tag{10}$$

where $\sigma_1, \ldots, \sigma_R$ are the diagonal entries of $\Sigma$, $u_j(x) = \sum_{s=1}^{R} U_{sj} Q_s^{\text{left}}(x)$, and $v_j(y) = \sum_{s=1}^{R} V_{sj} Q_s^{\text{right}}(y)$.

After we have formed eq. (10), we truncate the expansion down to $k < R$ terms to give a Mercer's expansion of $K$ truncated after $k$ terms. Then, this final approximation not only compresses the pseudo-skeleton approximant from step 1 and ensures that the resulting representation is very close to an optimal rank $\leq k$ approximation with respect to the $L^2$ norm.

If a rank-$R$ approximation of $K$ is computed and each slice needs to be evaluated $\mathcal{O}(n)$ times, then the cost of GECP and compression is about $\mathcal{O}(R^2 n + R^3)$ operations. If we discretize first and then compute the discrete SVD, the cost would be $\mathcal{O}(n^3)$ operations.

## 6    CONCLUSION

We derive new examples of kernels to show that the convergence properties of Mercer's expansion can be subtle for continuous kernels that are not symmetric positive definite. In particular, continuity alone is not enough to guarantee that a kernel has a Mercer's expansion, which holds pointwise. We then prove that any continuous kernel with uniform bounded variation has a Mercer's expansion that converges pointwise almost everywhere, unconditionally almost everywhere, and almost uniformly. We derive a new bound on the decay of singular values for smooth kernels and also provide an efficient algorithm for computing Mercer's expansion.

## 7    ACKNOWLEDGEMENTS

The second author was supported by the SciAI Center, funded by the Office of Naval Research under Grant Number N00014-23-1-2729 and NSF CAREER (DMS-2045646).

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

# A INDEFINITE KERNELS WHOSE MERCER'S EXPANSION CONVERGES POINTWISE BUT NOT ABSOLUTELY OR UNIFORMLY.

In this section, we prove that Mercer's expansion can converge pointwise, but not absolutely, and may converge pointwise, but not uniformly.

## A.1 A MERCER'S EXPANSION THAT CONVERGES POINTWISE, BUT NOT ABSOLUTELY

We begin by proving that for any $[a, b] \subset \mathbb{R}$ there exists a symmetric indefinite kernel $K_{\text{abs}} : [a, b] \times [a, b] \to \mathbb{R}$ with a Mercer's expansion that converges pointwise, but not absolutely.

*Proof of Proposition 2.* Without loss of generality, we assume that $[a, b] = [-1, 1]$; otherwise, we can transplant the example below to have this domain. Consider the kernel $K_{\text{abs}} : [-1, 1] \times [-1, 1] \to \mathbb{R}$ (also defined in eq. (6)):

$$K_{\text{abs}}(x, y) = \sum_{n=1}^{\infty} \frac{(-1)^n}{n^2} \tilde{P}_{2n}(x) \tilde{P}_{2n}(y), \tag{11}$$

where $\tilde{P}_n$ is the degree $n$ normalized Legendre polynomial. First, we show that the expansion in eq. (11) converges pointwise so that the value of $K_{\text{abs}}(x, y)$ is well-defined. Recall that Bernstein's inequality for $x \in (-1, 1)$, says that

$$|\tilde{P}_n(x)| < \sqrt{\frac{2}{\pi}} \frac{1}{(1 - x^2)^{1/4}}, \qquad n \geq 0.$$

Furthermore, $|\tilde{P}_n(x)| \leq \sqrt{n + 1/2}$ for all $x \in [-1, 1]$ (see (DLMF, 18.14.4) with $\lambda = 1/2$). Consider any closed set $S = [a, b] \times [-1, 1]$ with $-1 < a < b < 1$ and let $M = \sqrt{\frac{2}{\pi}} \max_{x \in \{a,b\}} \{(1 - x^2)^{-1/4}\}$. Then, $|\tilde{P}_n(y)| \leq M$ for any $n \geq 0$ and $y \in [a, b]$. Thus, for any $(x, y) \in S$, we have

$$\sum_{n=1}^{\infty} \left| \frac{(-1)^n}{n^2} \tilde{P}_{2n}(x) \tilde{P}_{2n}(y) \right| \leq M \sum_{n=1}^{\infty} \frac{\sqrt{n + 1/2}}{n^2} < \infty,$$

which, by the Weierstrass $M$-test, implies that the series in eq. (11) converges absolutely and uniformly on $[a, b] \times [-1, 1]$. Similarly, the series converges absolutely and uniformly on $[-1, 1] \times [a, b]$ for any $-1 < a < b < 1$. Consequently, we have pointwise convergence of the series in eq. (11) and continuity of $K$ everywhere except possibly at the four corners $(\pm 1, \pm 1)$. To check that the series converges pointwise at the four corners, note that $\tilde{P}_{2n}(\pm 1) = \sqrt{2n + 1/2}$ so

$$K_{\text{abs}}(\pm 1, \pm 1) = \sum_{n=1}^{\infty} \frac{(-1)^n}{n^2} \sqrt{2n + 1/2} \cdot \sqrt{2n + 1/2} = \sum_{n=1}^{\infty} (-1)^n \left( \frac{2}{n} + \frac{1}{2n^2} \right),$$

which is an alternating series that converges. Thus $K_{\text{abs}}(x, y)$ is well-defined pointwise by its expansion for any $(x, y) \in [-1, 1] \times [-1, 1]$.

To prove that $K_{\text{abs}}$ is continuous at $(\pm 1, \pm 1)$, we prove that $\hat{K}_{\text{abs}}(x, y) = \sum \frac{(-1)^n}{n} P_{2n}(x) P_{2n}(y)$ is continuous at the four corners. This suffices as $K_{\text{abs}}$ and $\hat{K}_{\text{abs}}$ only differ by a continuous function. Moreover, by symmetry, continuity at $(1, 1)$ guarantees continuity at all four corners.

Let us first prove the continuity of $\hat{K}_{\text{abs}}$ along the edge $[\frac{1}{2}, 1] \times \{1\}$, which will be useful. For any $(x, y)$ on $[\frac{1}{2}, 1] \times \{1\}$ we have

$$\hat{K}_{\text{abs}}(x, y) = \sum_{n=1}^{\infty} \frac{(-1)^n}{n} P_{2n}(x) =: G(x).$$

Using the generating function of Legendre polynomials (Szego, 1939), we find that

$$\frac{1}{t\sqrt{1 + t^2 - 2xt}} - \frac{1}{t} = \sum_{n=1}^{\infty} P_n(x) t^{n-1}, \qquad x \in [-1, 1].$$

Integrating both sides against $t$ from 0 to $s$ we see that

$$\int_0^s \left( \frac{1}{t\sqrt{1 + t^2 - 2xt}} - \frac{1}{t} \right) dt = \log \left( \frac{4(x - s + \sqrt{1 + s^2 - 2sx})}{(1 + x)(1 - s + \sqrt{1 + s^2 - 2sx})^2} \right) = \sum_{n=1}^{\infty} \frac{1}{n} t^n P_n(x).$$

We now do the substitution $s \to i$, multiply by 2, and take the real part, to obtain

$$\log \left( \frac{4}{(1 + \sqrt{x})^2 (1 + x)} \right) = \sum_{n=1}^{\infty} \frac{(-1)^n}{n} P_{2n}(x) = G(x), \tag{12}$$

for $x \in [\frac{1}{2}, 1]$. The left-hand side is a smooth function, which proves the continuity of $G(x)$ on $[\frac{1}{2}, 1]$. In particular, we have the continuity of $G(x)$ at $x = 1$.

Using the product formula for Legendre functions (Erdélyi et al., 1953, §3.15, eq. (20))

$$P_{2n}(x) P_{2n}(y) = \frac{1}{\pi} \int_0^{\pi} P_{2n} \left( xy + \cos\theta \sqrt{(1 - x^2)(1 - y^2)} \right) d\theta,$$

we have

$$\hat{K}_{\text{abs}}(x, y) = \sum_{n=1}^{\infty} \frac{(-1)^n}{n\pi} \int_0^{\pi} P_{2n} \left( xy + \cos\theta \sqrt{(1 - x^2)(1 - y^2)} \right) d\theta$$

$$= \frac{1}{\pi} \int_0^{\pi} G \left( xy + \cos\theta \sqrt{(1 - x^2)(1 - y^2)} \right) d\theta,$$

by eq. (12). (We can interchange the integration and summation due to uniform convergence of $G$ on $[a, b] \subset (-1, 1)$.) Note that $xy + \cos\theta \sqrt{(1 - x^2)(1 - y^2)}) \leq 1$ by the Cauchy–Schwarz inequality. From the mean value theorem and continuity of $G$, for any $x, y$ sufficiently close to 1, we have some $z \in [xy - \sqrt{(1 - x^2)(1 - y^2)}, xy + \sqrt{(1 - x^2)(1 - y^2)}]$ such that

$$\sum_{n=1}^{\infty} \frac{(-1)^n}{n} P_{2n}(x) P_{2n}(y) = G(z).$$

As $x, y \to 1$, the interval $\left[ xy - \sqrt{(1-x^2)(1-y^2)}, xy + \sqrt{(1-x^2)(1-y^2)} \right]$ shrinks down to 1 and $z \to 1$. By the smoothness of $G$, we conclude that $\sum_{n=1}^{\infty} \frac{(-1)^n}{n} P_{2n}(x) P_{2n}(y)$ is continuous at $(1, 1)$.

Finally to show that eq. (11) does not converge absolutely, consider the point $(x, y) = (1, 1)$. We have

$$\sum_{n=1}^{\infty} \left| \frac{(-1)^n}{n^2} \tilde{P}_{2n}(1) \tilde{P}_{2n}(1) \right| = \sum_{n=1}^{\infty} \frac{1}{n^2} \sqrt{2n + 1/2} \cdot \sqrt{2n + 1/2} = \sum_{n=1}^{\infty} \frac{2n + \frac{1}{2}}{n^2}$$

which does not converge. $\qquad\square$

The kernel $K_{\text{abs}}$ tells us that the convergence of Mercer's expansion can be subtle for general continuous kernels.

### A.2 A MERCER'S EXPANSION THAT CONVERGES POINTWISE, BUT NOT UNIFORMLY

We now prove that for any $[a, b] \subset \mathbb{R}$ there exists a symmetric indefinite kernel $K_{\text{uni}} : [a, b] \times [a, b] \to \mathbb{R}$ with a Mercer's expansion that converges pointwise, but not uniformly. This section contains a proof Proposition 3.

Without loss of generality, we assume that $[a, b] = [-1, 1]$; otherwise, we can transplant the example below to have this domain. First, define $\tilde{U}_n(x) := \sqrt{2/\pi}(1-x^2)^{1/4} U_n(x)$, where $U_n$ is the degree $n$ Chebyshev polynomial of the second kind. Note that $\{\tilde{U}_n\}_{n \in \mathbb{N}}$ is an orthonormal set of polynomials with respect to the standard $L^2$ inner-product on $[-1, 1]$. Also, $\tilde{U}_n(-1) = \tilde{U}_n(1) = 0$ and $\tilde{U}_n$ is continuous.

We now divide $[-1, 1]$ into infinite number of disjoint subintervals $I_1, I_2, \ldots$, defined by

$$I_n = \left( -1 + \frac{12}{\pi^2} \sum_{j=1}^{n-1} \frac{1}{j^2}, -1 + \frac{12}{\pi^2} \sum_{j=1}^{n} \frac{1}{j^2} \right),$$

which is designed so that $\text{length}(I_n) = \frac{12}{\pi^2 n^2}$, while $\sum_{n=1}^{\infty} \text{length}(I_n) = 2$. Using these subintervals, define $v_n$ for all $n \in \mathbb{N}$, as

$$v_n(x) = \begin{cases} -\dfrac{m\pi}{\sqrt{6}} \tilde{U}_{2m}(i_m(x)), & \text{if } n = 2m-1 \\ \dfrac{m\pi}{\sqrt{6}} \tilde{U}_{2m+2}(i_m(x)), & \text{if } n = 2m \end{cases} \tag{13}$$

where $i_m : I_m \to [-1, 1]$ is a linear bijection (shift) from $I_m$ onto $[-1, 1]$. The functions $v_1, v_2, \ldots$ are orthonormal on $L^2([-1, 1])$ and $v_n$ vanishes outside of $I_m$.

We now consider the kernel $K_{\text{uni}} : [-1, 1] \times [-1, 1] \to \mathbb{R}$ given by

$$K_{\text{uni}}(x, y) = \sum_{n=1}^{\infty} \frac{(-1)^n}{(\lceil \frac{n}{2} \rceil)^3} v_n(x) v_n(y). \tag{14}$$

The kernel $K_{\text{uni}}$ is continuous, symmetric, and has an infinite number of positive and an infinite number of negative eigenvalues. Also since $\{v_n\}_{n \in \mathbb{N}}$ is an orthonormal set, eq. (14) is a Mercer's expansion for $K_{\text{uni}}$. It turns out that this expansion does not converge uniformly to $K_{\text{uni}}$.

**Remark 1.** *The singular values of $K_{uni}$ in eq. (14) are $\frac{1}{1^3}, \frac{1}{1^3}, \frac{1}{2^3}, \frac{1}{2^3}, \ldots$, which have a decay rate of $\mathcal{O}(n^{-3})$ as $n \to \infty$.*

To prove Proposition 3, we introduce two useful lemmas about the functions $\tilde{U}_n$.

**Lemma 1** (Uniform norm bound of $\tilde{U}_n$)**.** *For any $n \in \mathbb{N}$ we have*

$$2\pi^{-1}\sqrt{n} \le \|\tilde{U}_n\|_{\infty} \le \sqrt{4(n+1)\pi^{-1}},$$

*where $\|\tilde{U}_n\|_{\infty}$ is the absolute maximum value of $\tilde{U}_n(x)$ for $x \in [-1, 1]$.*

*Proof.* With the change of variables $x = \cos\theta$ for $\theta \in [0, \pi]$, we find that

$$\sqrt{\frac{\pi}{2}} \tilde{U}_n(x) = \frac{\sin(n+1)\theta}{\sqrt{\sin\theta}}.$$

This means that we have

$$\sqrt{\frac{\pi}{2}} \tilde{U}_n(x_n) = \frac{\sin\frac{\pi}{2}}{\sqrt{\sin\frac{\pi}{2(n+1)}}} > \sqrt{\frac{2}{\pi}(n+1)} > \sqrt{\frac{2n}{\pi}},$$

for $x_n = \cos\frac{\pi}{2(n+1)}$ using $\sin(x_n) < x_n$. This proves the lower bound on $\|\tilde{U}_n\|\infty$.

To prove the upper bound, first consider $0 \le \theta < \frac{1}{n+1}$ to see that

$$\frac{\sin(n+1)\theta}{\sqrt{\sin\theta}} \le \frac{(n+1)\theta}{\sqrt{\theta/2}} < \sqrt{2\theta}(n+1) < \sqrt{2(n+1)},$$

using $\frac{\theta}{2} \le \sin\theta$ for $\theta \in [0, \pi/2]$. For $\frac{1}{n+1} \le \theta \le \frac{\pi}{2}$, we have

$$\frac{\sin(n+1)\theta}{\sqrt{\sin\theta}} \le \frac{1}{\sqrt{\sin\theta}} \le \sqrt{\frac{2}{\theta}} \le \sqrt{2(n+1)}.$$

Finally, the upper bound on $\|\tilde{U}\|_\infty$ holds since $|\tilde{U}_n(-x)| = |\tilde{U}_n(x)|$. $\qquad\square$

Lemma 1 is a weaker version of Bernstein-type bound for Chebyshev polynomials of the second kind. Recall that the Chebyshev polynomials of the second kind are the Jacobi polynomials with parameter $(\frac{1}{2}, \frac{1}{2})$. For Jacobi polynomials with integer parameters, a similar bound is known (Haagerup & Schlichtkrull, 2014), while it is discussed only numerically so far for Jacobi polynomials with noninteger parameters (Koornwinder et al., 2018, Remark 4.4).

**Lemma 2** (Sum of Consecutive $(-1)^n\tilde{U}_{2n}$)**.** *For any $n \in \mathbb{N}$, we have*

$$\left\| \tilde{U}_{2n}(x) - \tilde{U}_{2n+2}(x) \right\|_\infty \le \sqrt{8\pi^{-1}}.$$

*Proof.* Using the trigonometric definition of $\tilde{U}_n$, we have

$$\sqrt{\frac{\pi}{2}} \left( \tilde{U}_{2m}(x) - \tilde{U}_{2m+2}(x) \right) = \frac{\sin((2m+1)\theta) - \sin((2m+3)\theta)}{\sqrt{\sin\theta}}$$
$$= -2\cos((2m+2)\theta)\sqrt{\sin\theta},$$

which gives the desired inequality. $\qquad\square$

Lemma 2 shows us that if one adds two consecutive terms from $\{(-1)^n\tilde{U}_{2n}(x)\}_{n\in\mathbb{N}}$, the norm of the resulting function is bounded by a constant, whereas Lemma 1 states that each term on its own has norm at least $\mathcal{O}(\sqrt{n})$. In other words, there will be significant pointwise cancellations happening everywhere in Mercer's expansion for $K_{\text{uni}}$ in eq. (14).

We are ready to prove Proposition 3 by giving an explicit example of symmetric indefinite kernel with pointwise convergent Mercer's expansion that does not converge uniformly.

*Proof of Proposition 3.* First, notice that from the definition of $v_n$ and $K_{\text{uni}}(x, y)$, there are at most two nonzero terms in the series for each fixed $x, y \in [-1, 1]^2$. If $m$ is an integer such that $x \in I_m$, then $K_{\text{uni}}(x, y) = 0$ if $y \notin I_m$; otherwise, for $y \in I_m$, we find that

$$K_{\text{uni}}(x, y) = \frac{1}{m^3}(v_{2m-1}(x)v_{2m-1}(y) + v_{2m}(x)v_{2m}(y))$$
$$= \frac{1}{m^3}\frac{m^2\pi^2}{6}\left[ \tilde{U}_{2m+2}(i_m(x))\tilde{U}_{2m+2}(i_m(y)) - \tilde{U}_{2m}(i_m(x))\tilde{U}_{2m}(i_m(y)) \right], \quad (15)$$

which converges for every $x, y$. For any $x, y$ in $([-1, 1] \times [-1, 1]) \backslash ([b, 1] \times [b, 1])$ with $b < 1$ the expansion in eq. (14) is a finite sum, which converges uniformly and $K_{\text{uni}}$ is continuous on $[-1, 1] \times [-1, 1]$ except possibly at the point $(x, y) = (1, 1)$ as the functions $\{v_n\}_{n \in \mathbb{N}}$ are continuous.

To prove continuity at the upper right corner $(1, 1)$, consider a small open set $B$ around $(1, 1)$. For any $x, y \in B$ suppose $x, y \in I_m$. Letting $x' = i_m(x), y' = i_m(y)$ and from eq. (15) we have

$$K_{\text{uni}}(x, y) = \frac{\pi^2}{6m} \left[ \left( \tilde{U}_{2m+2}(x') - \tilde{U}_{2m}(x') \right) \tilde{U}_{2m+2}(y') \right.$$
$$\left. + \tilde{U}_{2m}(x') \left( \tilde{U}_{2m+2}(y') - \tilde{U}_{2m}(y') \right) \right].$$

By Lemma 2 the terms $\tilde{U}_{2m+2}(x') - \tilde{U}_{2m}(x')$ and $\tilde{U}_{2m+2}(y') - \tilde{U}_{2m}(y')$ are bounded above by $2\sqrt{\frac{2}{\pi}}$, and $\tilde{U}_{2m+2}, \tilde{U}_{2m}$ are both bounded above by $\sqrt{(8m + 12)/\pi}$. Thus we obtain

$$|K_{\text{uni}}(x, y)| \leq \frac{\pi^2}{6m} \cdot 2\sqrt{\frac{2}{\pi}} \cdot 2\sqrt{\frac{8m + 12}{\pi}} < Cm^{-1/2},$$

where $C$ does not depend on $m$. As $B$ gets smaller, we have $x, y \to 1$, which implies $m \to \infty$. Hence, limit of the value of $K_{\text{uni}}(x, y)$ as $(x, y) \to (1, 1)$ is zero, which equal to $K_{\text{uni}}(1, 1)$, proving the continuity of $K_{\text{uni}}$ at $(1, 1)$. Altogether, we find that $K_{\text{uni}}$ is continuous on $[-1, 1] \times [-1, 1]$.

For the final step, let us prove that the convergence of eq. (16) is not uniform. The norm of $(2m)$th term of the right-hand side of eq. (14) is bounded below by

$$\frac{1}{m^3} \max_{x, y \in [-1, 1]} |v_{2m}(x) v_{2m}(y)| > \frac{1}{m^3} \cdot \frac{m^2 \pi^2}{6} \cdot \left( \frac{2}{\pi} \sqrt{2m} \right)^2 = \frac{4}{3}$$

using the lower bound of $\tilde{U}_{2m+2}$ obtained in Lemma 1. Since this constant lower bound holds for any $(2m)$th term, it can be deduced from the Cauchy condition that the series does not converge uniformly. $\qquad \square$

The following proposition gives an additional example of a continuous asymmetric kernel where its SVE converges pointwise but not uniformly. Once more, we use the functions $\{v_n\}$ defined in eq. (13).

**Proposition 5** (Asymmetric kernel with a Mercer's expansion that does not converge uniformly). *Define $K_{as} : [-1, 1] \times [-1, 1] \to \mathbb{R}$ by*

$$K_{as}(x, y) = \sum_{n=1}^{\infty} \frac{1}{(2 \lceil \frac{n}{2} \rceil)^2} u_n(x) v_n(y), \tag{16}$$

*with $u_n = \tilde{U}_{2n}$ and $v_n$ defined in eq. (13). Then, $K_{as}$ is a continuous function which is well-defined for all $(x, y) \in [-1, 1] \times [-1, 1]$. Moreover, the right-hand side of eq. (16) is Mercer's expansion of $K_{as}$ which does not converge uniformly.*

We omit the proof as it is analogous to that of Proposition 3.

## B ON THE CONVERGENCE OF MERCER'S EXPANSION FOR CONTINUOUS KERNELS OF UNIFORM BOUNDED VARIATION

For any $[a, b], [c, d] \subset \mathbb{R}$ and a continuous kernel $K : [a, b] \times [c, d] \to \mathbb{R}$ with uniform bounded variation (see eq. (3)), we prove that Mercer's expansion of $K$ converges unconditionally almost everywhere.

To show this, we first give the following lemma, which is a generalization of the Rademacher–Menchov Theorem.

**Lemma 3** (Kashin & Saakyan (2005), Chapter VIII.2). *Let $\{\phi_n(x)\}_{n \in \mathbb{N}}$ be a set of orthonormal functions and $\{a_n\}_{n \in \mathbb{N}}$ be a set of coefficients. If $a_1, a_2, \dots$ satisfy $\sum_{n=1}^{\infty} |a_n|^{2-\epsilon} < \infty$ for some $\epsilon > 0$ then the series $\sum_{n=1}^{\infty} a_n \phi_n(x)$ converges unconditionally almost everywhere.*

Lemma 3 allows us to prove that Mercer's expansion, i.e.,

$$K(x, y) = \sum_{n=1}^{\infty} \sigma_n u_n(x) v_n(y),$$

converges unconditionally almost everywhere for continuous kernels of uniform bounded variation.

*Proof of Theorem 2.* Let $\epsilon > 0$. By Hölder's inequality the following inequality holds for any $\delta > 0$:

$$\sum_{n=1}^{\infty} |\sigma_n u_n(x)|^{2-\epsilon} = \sum_{n=1}^{\infty} |\sigma_n|^{\delta} \left( |\sigma_n|^{2-\epsilon-\delta} |u_n(x)|^{2-\epsilon} \right)$$

$$\leq \left( \sum_{n=1}^{\infty} \left( |\sigma_n|^{\delta} \right)^{\frac{2}{\epsilon}} \right)^{\frac{\epsilon}{2}} \cdot \left( \sum_{n=1}^{\infty} \left( |\sigma_n|^{2-\epsilon-\delta} |u_n(x)|^{2-\epsilon} \right)^{\frac{2}{2-\epsilon}} \right)^{\frac{2-\epsilon}{2}}, \qquad (17)$$

The first term $(\sum_{n=1}^{\infty} |\sigma_n|^{\frac{2\delta}{\epsilon}})^{\frac{\epsilon}{2}}$ converges when

$$\alpha \cdot \frac{2\delta}{\epsilon} > 1 \qquad (18)$$

while the second term can be written as

$$\left( \sum_{n=1}^{\infty} \left( |\sigma_n|^{2-\epsilon-\delta} |u_n(x)|^{2-\epsilon} \right)^{\frac{2}{2-\epsilon}} \right)^{\frac{2-\epsilon}{2}} = \left( \sum_{n=1}^{\infty} |\sigma_n|^{\frac{4-2\epsilon-2\delta}{2-\epsilon}} |u_n(x)|^2 \right)^{\frac{2-\epsilon}{2}}.$$

Integrating the sum inside the parenthesis with respect to $x$, we find that

$$\int_0^1 \sum_{n=1}^{\infty} |\sigma_n|^{\frac{4-2\epsilon-2\delta}{2-\epsilon}} |u_n(x)|^2 dx = \sum_{n=1}^{\infty} \left( \int_0^1 |\sigma_n|^{\frac{4-2\epsilon-2\delta}{2-\epsilon}} |u_n(x)|^2 dx \right) = \sum_{n=1}^{\infty} |\sigma_n|^{\frac{4-2\epsilon-2\delta}{2-\epsilon}},$$

provided that the right-hand side converges, i.e.,

$$\alpha \cdot \frac{4 - 2\epsilon - 2\delta}{2 - \epsilon} > 1. \qquad (19)$$

In this case, we have a monotonic series of functions given by

$$f_n(x) = \sum_{n=1}^{n} |\sigma_n|^{\frac{4-2\epsilon-2\delta}{2-\epsilon}} |u_n(x)|^2,$$

which converges in the $L^1$ sense. Since we have $L^1$ convergence, we know that the second term of eq. (17) converges pointwise for almost every $x$.

From the assumption that we have $\alpha > \frac{1}{2}$, the two conditions in eq. (18) and eq. (19) are both satisfied by selecting $\delta, \epsilon$ such that $\delta > \epsilon > 0$ and

$$1 - \frac{1}{2\alpha} > \frac{\delta}{2 - \epsilon}.$$

(There is always a solution when $\alpha > \frac{1}{2}$ as one can take $\delta = 1 - \frac{1}{2\alpha}$ and $\epsilon = \min\{0.001, \delta/2\}$.) Thus, for any given $\alpha > \frac{1}{2}$, there is $\delta, \epsilon > 0$ such that eq. (17) converges for almost every $x$. From Lemma 3 we see that $\sum_{n=1}^{\infty} \sigma_n u_n(x) v_n(y)$ converges unconditionally for almost every $x$ and almost every $y$. $\qquad \square$

