# OpenReview forum: "Extending Mercer's expansion to indefinite and asymmetric kernels"
_ICLR.cc/2025/Conference — ICLR 2025 Poster_

### Official Review · Reviewer_5Zo5 · 2024-10-18

**Soundness:** 3
**Presentation:** 3
**Contribution:** 2
**Rating:** 8
**Confidence:** 4

**Summary:**

This paper shows that if an asymmetric and non-positive definite kernel satisfies the uniform bounded variation property on the two coordinates (Eq. (3)), then it admits a Mercer decomposition which converges pointwise, unconditionally almost everywhere, and almost uniformly. Also, the paper provides an algorithm on computing Mercer expansion for general kernels.

**Strengths:**

The paper is written in a rigorous manner where the proof of the main theorem is stated clearly in the appendix. Also, the authors explain the importance of this paper as filling the gap in the literature on general asymmetric and non-positive definite kernels.

**Weaknesses:**

However, the contribution of the paper seems limited. The kernel needs to be defined on two (different) intervals in the theorem, which is too simple for practical implications. And the proof of the main theorem is more or less the result from Rademacher–Menchov Theorem and Hölder inequality. I think that the authors could have extended the proof to kernels with high-dimensional compact input spaces, or explain why it is difficult to perform such an extension.

**Questions:**

1. Could the authors explain why they did not extend the main theorem to kernels with high-dimensional compact input spaces?
2. I am not very familiar with but interested in computing/approximating Mercer extension of a general kernel. Is the procedure in section 5 novel or is it a standard approach in the literature? If it is novel, it is worth to explain more on the algorithm's complexity and accuracy.
3. In line 639 "We believe the bounds in eq. (8) are asymptotically tight." Do you have any experimental results with general asymmetric kernels to support your hypothesis?

---

> ### Author Response · Authors · 2024-11-19
> **Response to the reviewer 5Zo5**
>
> We thank the referee for careful and thorough review. We also appreciate valuable and insightful questions raised by the reviewer.
>
> To answer the reviewer's first question about high-dimensional input spaces, we first point out that our counterexamples in Proposition 2 and 3 still work as a representative case for all compact domains. However, our convergence results in Section 3.4, do not currently apply to general high-dimensional domains. The two main ingredients in our proof (decay of Legendre coefficients and convergence of orthonormal series) are extremely understudied for dimension $n>1$, e.g., even for the simplest case of $[0, 1]^2$. In fact, this leads to another interesting remark, that Mercer's expansion is very special because it is insensitive to the underlying topology of the domain; thus, can be easily extended to general compact domains. Intuitively, this is due to the effect of positive-definiteness, which prevents terms in the series canceling out each other. We have added a short discussion on general domains in the revised manuscript.
>
> We also want to discuss the question regarding the algorithm for computing our generalized Mercer's expansion. The algorithm presented in Section 5 of our manuscript is described in [1] (in particular, see Figure 6.1 of [1] for an algorithmic summary), which we referenced in the manuscript. The algorithm's complexity is also discussed in their work. The underlying idea of pseudo-skeleton approximation traces back to the foundational work by Goreinov, Tyrtyshnikov, and Zamarashkin [2] (also referenced in manuscript). While the algorithm itself is not new, we chose to include it in this manuscript because we believe it is a novel approach for computing Mercer’s expansion in the machine learning literature. By incorporating it here, we hope to bring attention to its utility and potential applications for computing Mercer's expansions. We have updated the manuscript to make sure that a reader knows that we have a novel application of an existing algorithm.
>
> Moreover, while deriving the decay bounds on the singular values in our manuscript, we conducted numerous numerical experiments on functions with varying degrees of smoothness. This is the basis for our belief that the bounds in Eq. (8) are asymptotically tight, and we are satisfied with the results from this perspective. However, we acknowledge that we have not been able to theoretically prove that improved bounds are unattainable. We have updated the manuscript to refer to a figure from [3] (Figure 2.4) that showcases three numerical experiments. These experiments support the hypothesis that the bounds are indeed asymptotically tight. We hope this addition clarifies the reasoning behind our assertion and provides further evidence for the referee to evaluate.
>
> Thank you again and please let us know if you have further questions.
>
>
> [1] A. Townsend and L. Trefethen, An Extension of Chebfun to Two Dimensions. SIAM Journal
> on Scientific Computing, 35(6):C495–C518, 2013.
>
> [2] S. Goreinov, E. Tyrtyshnikov, and N. Zamarashkin, A theory of pseudoskeleton approximations. Linear Algebra and its Applications, 261(1-3):1–21, 1997.
>
> [3] A. Townsend, Computing with functions in two dimensions. PhD thesis, Oxford University, UK,
> 2014.

---

> > ### Comment · Reviewer_5Zo5 · 2024-11-20
> >
> > Thank you for your answer. I raise my score under the condition that the authors would add more discussion to their papers as mentioned in their reply.

---

### Official Review · Reviewer_nauL · 2024-11-01

**Soundness:** 3
**Presentation:** 3
**Contribution:** 2
**Rating:** 6
**Confidence:** 3

**Summary:**

This manuscript studied the Mecer's decomposition for indefinite and asymmetric kernel.

They extend Mercer's expansion to continuous kernels, providing some theoretical underpinning for indefinite and asymmetric kernels.

1. Mercer's expansion may not be pointwise convergent for continuous indefinite kernels,
2. Prove that the expansion of continuous kernels with bounded variation uniformly in each variable separably converges pointwise almost everywhere, almost uniformly, and unconditionally almost everywhere.
3. Describe an algorithm for computing Mercer's expansion for general kernels and give new decay bounds on its terms.

**Strengths:**

The authors claim to have established some fundamental results for ``Mercer's decomposition'' for indefinite, asymmetric kernels.

The following points seem novel to me:

1. It is generally expected that ``Mercer's decomposition'' does not behave well when the kernel \( K \) is not positive definite and asymmetric. The authors provide several examples of this behavior:
   - 1. It does not converge pointwise.
   - 2. It converges pointwise but not absolutely.
   - 3. It converges pointwise but not uniformly.

2. They demonstrate that if the decay rate of the singular values is sufficiently fast and there is some smoothness condition on the kernel, there are some unconditional convergence results.

3. They assert that the smoother the kernel, the faster the decay rate of the singular values.

**Weaknesses:**

The concern regarding the mathematical novelty of this paper is that most of the results in this manuscript are anticipated.

Since it is not fair to judge an ICLR paper solely based on its mathematical novelty, it would be beneficial if the authors could explore the necessity of studying Mercer's decomposition for indefinite, asymmetric kernels more thoroughly.

1.Could the authors discuss why the utilization of asymmetric kernels is necessary in data analysis?
2.Could the authors provide examples of several classes of asymmetric kernels of general interest?
3.Are there any application examples that demonstrate the use of asymmetric kernels?

**Questions:**

Same as the weakness. Especially the application consideration of the asymmetric kernel/

---

> ### Author Response · Authors · 2024-11-19
> **Response to the reviewer nauL**
>
> We greatly thank the reviewer for a thorough review and valuable comments.
>
> **Regarding the reviewer's comment "The concern regarding the mathematical novelty of this paper is that most of the results in this manuscript are anticipated,"** we think our results are unanticipated. Mercer's expansion for general continuous kernels has been anticipated to behave nicely; however, we are showing that it doesn't without additional smoothness assumptions. There are several examples in the literature of erroneous statements regarding general kernels and their Mercer's expansions, including claims that Mercer's theorem continues to hold for general continuous kernels. We have shown this is not true.
>
> There are many erroneous statements in the literature that demonstrate our results are unanticipated:
>
> - For example, in [1], the authors incorrectly proves that any continuous indefinite kernel has a convergent Mercer's expansion. They assume that for any indefinite kernel the positive and negative parts can be separated; however, this is not true in the pointwise sense, and the kernel $K_\text{abs}$ in Section 3.3 of our manuscript is a counterexample.
> - For another example, in Chapter 3 of [2] the authors argue that any continuous kernel on a compact domain has a pointwise convergent SVE, by using the Stone-Weierstrass Theorem. This is incorrect due to our Proposition 1 where our example in the proof serves as a direct counterexample. (Their argument only proves that a kernel is a limit of a series of finite rank kernels, which may have non-converging series.)
> - Another recent publication on the self-attention of Transformers [3], claims that a general continuous kernel is equal to the inner product (Definition 2.1) of feature spaces. This is also incorrect as we provided an example which shows that Eq.(4) of [3] does not even hold pointwise. To be more precise, such an infinite sum (inner product on Hilbert or Banach space) should be treated with an extreme care and the equality (which is, in fact, a convergence) always has to be further clarified.
> - In [11], the authors propose techniques to modify any indefinite kernels by manipulating Mercer's expansion, which is actually impossible for many indefinite kernels. The infinite series has to be carefully examined because it may not converge as we have seen in our manuscript.
> - Lastly, in [4], the author makes a mistake, stating that for continuous (uniformly) Lipschitz kernels Mercer's expansion converges absolutely. This also has to be corrected to absolute convergence almost everywhere.
>
> We gently point some of these errors out in our manuscript, showing that our results are unanticipated.
>
> &nbsp;
> &nbsp;
> &nbsp;
> &nbsp;
>
> **Regarding your comment on applications and necessity of studying Mercer's expansion for general kernels**, in the third paragraph of the introduction we discuss a few reasons why indefinite and asymmetric kernels are necessary. We have summarized three reasons there. Similar and further reasons are illustrated, mainly in [7] and in few other articles we cited.
>
> Moreover, in the third paragraph of the introduction we referenced some work which discuss the role and appearance of asymmetric and indefinite kernels in the application of machine learning. To list some of them:
>
> - Most recently they arise in the context of self-attention of Transformers [3]. In the work by Wright and Gonzalez [6] the authors write an extensive list (see the third paragraph of the introduction and Section 2.2.2 of [6]) of publications concerning indefinite and asymmetric kernels, in addition to Transformers.
> - In $H^\infty$ control theory and applications [8].
> - Biology applications, e.g., Homology detection in protein sequences [9].
>
> To give some explicit examples of indefinite and asymmetric kernels, sigmoid ($\tanh$) kernel can appear as the activation function in neural networks. Local alignment kernel appears in biological applications [9], and furthermore BLAST and Smith-Waterman algorithms (also protein sequence similarity measurement) give arise to indefinite kernels. Moreover there are few other explicit examples illustrated in Table 1 of [7].
>
> We revised our manuscript so that it discusses more applications with details.
>
> Thank you again for the questions and comments, and please let us know if you have further questions!

---

> > ### Comment · Reviewer_nauL · 2024-11-26
> >
> > Thank you for your response. I would raise my score and lean toward accepting this paper.

---

> ### Author Response · Authors · 2024-11-19
> **References for the previous comment by the authors**
>
> [1] D. Chen, H. Wang, and E. Tsang, Generalized Mercer theorem and its application to feature space related to indefinite kernels, In 2008 International Conference on Machine Learning and Cybernetics, volume 2, pp. 774–777. IEEE, 2008.
>
> [2] Y. Xu and Q. Ye, Generalized Mercer kernels and reproducing kernel Banach spaces, volume 258. American Mathematical Society, 2019.
>
> [3] Y. Chen, et al., Primal-attention: Self-attention through asymmetric kernel svd in primal representation. Advances in Neural Information Processing Systems, 2024.
>
> [4] A. Townsend, Computing with functions in two dimensions. PhD thesis, Oxford University, UK, 2014.
>
> [5] M. He, et al., Learning with asymmetric kernels: Least squares and feature interpretation, IEEE Transactions on Pattern Analysis and Machine Intelligence 45.8 (2023): 10044-10054.
>
> [6] M. Wright and J. Gonzalez, Transformers are deep infinite-dimensional non-Mercer binary kernel machines, arXiv preprint arXiv:2106.01506, 2021.
>
> [7] C. Ong, X. Mary, S. Canu, and A. Smola, Learning with non-positive kernels. In Proceedings of the Twenty-first International Conference on Machine learning, pp. 81, 2004.
>
> [8] B. Hassibi,  A. Sayed, and T. Kailath. Indefinite-Quadratic estimation and control: a unified approach to $H^2$ and $H^\infty$ theories. Society for Industrial and Applied Mathematics, 1999.
>
> [9] J-P. Vert, H. Saigo, and T. Akutsu. Local alignment kernels for biological sequences, Kernel methods in computational biology, 2004.
>
> [10] R. Luss and A. d’Aspremont. Support vector machine classification with indefinite kernels. Advances in Neural Information Processing Systems, 20, 2007.
>
> [11] G. Wu, E. Chang, and Z. Zhang. An analysis of transformation on non-positive semidefinite similarity matrix for kernel machines. In Proceedings of the 22nd International Conference on Machine Learning, volume 8. Citeseer, 2005.

---

### Official Review · Reviewer_VyrL · 2024-11-07

**Soundness:** 3
**Presentation:** 3
**Contribution:** 3
**Rating:** 6
**Confidence:** 3

**Summary:**

This paper extended  Mercer’s theorem to any continuous kernels with bounded range, removing the original requirement for the kernel function to be symmetric and positive definite. Two main results are derived in this paper: first, it is proven that Mercer's expansion may not converge absolutely and uniformly for indefinite kernel; second, it is proven that a kernel with uniform bounded variation has Mercer's expansion that coverges pointwise a.e., a.u. and unconditionally a.e.. Notably, the authors described an algorithm for computing Mercer's expansion for general kernels.

**Strengths:**

1. The theoretical analysis in this paper is comprehensive and rigorous.

2. The paper establishes a theoretical foundation for Mercer’s expansion applied to non-regular kernels, such as indefinite and asymmetric kernels, potentially offering fundamental tools for future research in the theory of kernel-based method.

3.  A sufficient condition of uniformly bounded variation is proposed to ensure the validity of Mercer’s expansion.

**Weaknesses:**

The parameter $\alpha$  first appears in Theorem 2 (maybe I missed something)  before it is introduced (in Line 303). I suggest the authors make the writing more self-contained.

**Questions:**

The present results apply only to kernel functions defined on product intervals, specifically  $K: [a,b]\times [c,d]\to R$.  My question is whether by following a similar argument as that in your manuscript, these results could be extended to more general settings, such as compact domains or multiple dimensions.

---

> ### Author Response · Authors · 2024-11-19
> **Response to the reviewer VyrL**
>
> We greatly appreciate the careful review and valuable comments by the reviewer.
>
> Regarding the ambiguity of the parameter $\alpha$, we have revised our manuscript and the Holder exponent $\alpha$ in lines 301-303 is now changed to $\gamma$ to avoid confusion. Also, Theorem 2 has been very slightly revised for clarity.
>
> To answer your question about general domains, we first point out that our counterexamples in Proposition 2 and 3 still work as a representative case for all compact domains. However, our convergence results in Section 3.4, do not currently apply to general domains. The two main ingredients in our proof (decay of Legendre coefficients and convergence of orthonormal series) are extremely understudied for general domains, e.g., even for the simplest case of $[0, 1]^2$. This leads to another interesting remark, that Mercer's expansion is very special because it is insensitive to the underlying topology of the domain; thus, can be easily extended to general compact domains. Intuitively, this is due to the effect of positive-definiteness, which prevents terms in the series canceling out each other. We have added a short discussion on general domains in the revised manuscript.
>
> We hope this resolves all the questions by the reviewer. Please let us know if you have any more questions!

---

> > ### Comment · Reviewer_VyrL · 2024-11-24
> >
> > Thank you for your response and for providing new insights. I would keep my score and lean toward accepting this paper.

---

### Meta-Review · Area_Chair_Yt8E · 2024-12-20

**Metareview:**

The paper investigates Mercer's expansion for continuous kernels in the settings of indefinite and asymmetric kernels. In particular, the authors demonstrate that Mercer's expansion may not be pointwise convergent for continuous indefinite kernels. They then prove that the expansion of continuous kernels with bounded variation uniformly in each variable separably converges pointwise almost everywhere, almost uniformly, and unconditionally almost everywhere.

Reviewers agree that the theoretical analysis is rigorous and makes a solid contribution to the theoretical foundation of indefinite and asymmetric kernels.

**Additional Comments On Reviewer Discussion:**

- Reviewer VyrL,  5Zo5: the presented results apply to kernels on the product interval $[a,b] \times [c,d] \rightarrow \mathbb{R}$ and not a
general compact metric space.

The authors pointed out that their counterexamples are still valid, which also disprove several claims made recently in the literature.

On the other hand, I agree that the generalization to compact metric spaces of the convergence results would be much more valuable. Having said that, I believe that the current results are of sufficient interest to recommend acceptance.

---

### Decision · Program_Chairs · 2025-01-22

Accept (Poster)